# Spatial Patterns of Endometriosis Incidence. A Study in Friuli Venezia Giulia (Italy) in the Period 2004–2017

**DOI:** 10.3390/ijerph18137175

**Published:** 2021-07-05

**Authors:** Dolores Catelan, Manuela Giangreco, Annibale Biggeri, Fabio Barbone, Lorenzo Monasta, Giuseppe Ricci, Federico Romano, Valentina Rosolen, Gabriella Zito, Luca Ronfani

**Affiliations:** 1Unit of Biostatistics, Epidemiology and Public Health Department of Cardiac, Thoracic, Vascular Sciences and Public Health, University of Padova, 35121 Padova, Italy; dolores.catelan@unipd.it; 2Institute for Maternal and Child Health-IRCCS “Burlo Garofolo”, 34137 Trieste, Italy; lorenzo.monasta@burlo.trieste.it (L.M.); giuseppe.ricci@burlo.trieste.it (G.R.); federico.romano@burlo.trieste.it (F.R.); valentina.rosolen@burlo.trieste.it (V.R.); gabriella.zito@burlo.trieste.it (G.Z.); luca.ronfani@burlo.trieste.it (L.R.); 3Department of Statistics, Computer Science, Applications ‘G. Parenti’ (DiSIA), University of Florence, 50134 Firenze, Italy; annibale.biggeri@unifi.it; 4Department of Medical Area, University of Udine, 33100 Udine, Italy; fabio.barbone@uniud.it; 5Department of Medical, Surgical and Health Sciences, University of Trieste, 34149 Trieste, Italy

**Keywords:** endometriosis, Incidence, Epidemiological surveillance, disease mapping, hierarchical Bayesian models, High Risk Areas Profiling

## Abstract

Background: Diagnosis of endometriosis and evaluation of incidence data are complex tasks because the disease is identified laparoscopically and confirmed histologically. Incidence estimates reported in literature are widely inconsistent, presumably reflecting geographical variability of risk and the difficulty of obtaining reliable data. Methods: We retrieved incident cases of endometriosis in women aged 15–50 years using hospital discharge records and pathology databases of the Friuli Venezia Giulia region in the calendar period 2004–2017. We studied the spatial pattern of endometriosis incidence applying Bayesian approaches to Disease Mapping, and profiled municipalities at higher risk controlling for multiple comparisons using both q-values and a fully Bayesian approach. Results: 4125 new cases of endometriosis were identified in the age range 15 to 50 years in the period 2004–2017. The incidence rate (x100 000) is 111 (95% CI 110–112), with a maximum of 160 in the age group 31–35 years. The geographical distribution of endometriosis incidence showed a very strong north-south spatial gradient. We consistently identified a group of five neighboring municipalities at higher risk (RR 1.31 95% CI 1.13; 1.52), even accounting for ascertainment bias. Conclusions: The cluster of 5 municipalities in the industrialized and polluted south-east part of the region is suggestive. However, due to the ecologic nature of the present study, information on the patients’ characteristics and exposure histories are limited. Individual studies, including biomonitoring, and life-course studies are necessary to better evaluate our findings.

## 1. Introduction

Endometriosis is an estrogen-dependent female chronic inflammatory disease in which endometrial tissue develops outside the uterus. [1,2,3] The symptoms of endometriosis are chronic pelvic pain, fatigue, dysmenorrhea, dyspareunia, abnormal/irregular uterine bleeding and infertility or subfertility, as response of endometrial tissue to hormonal stimulation [4,5].

Endometriosis causes high social and healthcare system costs, worsening quality of life and work productivity. The average time from onset of symptoms to diagnosis ranges from 6 to 12 years, delaying appropriate therapy [6,7].

Diagnosing endometriosis also depends on the gynecologist and the assessment method used. Identifying cases in the general population from endometriosis registers is difficult because laparoscopic visualizations and histologic confirmation can only obtain a definitive diagnosis of the disease according to the ESHRE (European Society of Human Reproduction and Embryology) guidelines [8]. Laparoscopy is an invasive surgical procedure that is indicated in women with clear symptoms [9,10]. However, many women are asymptomatic, or there may be an overlap of symptoms with other conditions and lesions that might heal spontaneously or following hormonal treatment without a previously made diagnosis [11,12].

The estimated incidence of endometriosis reported in the literature shows considerable variability. It is difficult to interpret, reflecting geographical variability of risk or simply due to the difficulty mentioned above of obtaining reliable data [4,11,13,14,15,16,17].

In a previous paper [18], we estimated the incidence and prevalence of endometriosis in women between 15–50 years, in the Friuli Venezia Giulia (FVG) region, in the period 2011–2013. We found an endometriosis incidence rate of 0.11% and a prevalence of 1.82%.

When risk factors for disease are unknown, maps may help highlight spatial trends in the distribution of relative risk and generate etiological hypotheses.

However, when the inferential goal is to identify municipalities with risk diverging from the local or national reference, the problem must be addressed with a multiple test approach, testing hypothesis departure from the reference null for each municipality.

In public health implications of disease mapping, several different decision rules are discussed [19]. However, in practice, despite some criticism, *p*-values are still used to scrutinize long lists of relative risks.

In functional Genomics data analysis and other high-throughput biological areas of application, control of the False Discovery Rate (FDR) has become popular for multiple comparisons adjustment [20,21]. To date, limited research has been conducted on FDR in disease mapping [22].

In the Bayesian context, posterior probabilities of RR greater than the null obtained from hierarchical Bayesian models failed to address the multiple comparison problem [23,24,25]. As an alternative, classification probabilities obtained from Bayesian mixture models have been adjusted for multiple comparisons [26] and proposed in the context of disease mapping [27].

The goals of our study were to update the results reported in our previous paper [18]. We also sought to describe the spatial pattern of the incidence of endometriosis in FVG in the period 2004–2017 using the two most common Bayesian approaches of Disease Mapping. We also aimed to identify municipalities at different risk of endometriosis, controlling for multiple comparisons using both “frequentist” FDR and a fully Bayesian approach.

## 2. Materials

### 2.1. Incidence of Endometriosis

Data were extracted from the FVG Regional Repository of MicroData (RRMD) by record linkage using a unique anonymous identifier. RRMD is a centralized record system automatically pooling health care data from the national health service. The FVG Regional Health Authority granted access to the data, so no approval was required from the institutional review board of our Institute. FVG is a region located in the northeast of Italy. It covers 7855 Km^2^ and has a population of approximately 1.22 million people, of which 630 thousand are women.

All endometriosis cases were extracted from hospital discharge and anatomic pathology databases for the years 2004–2017. We used the International Classification of Diseases, Ninth Revision ICD-9, codes 617.0–617.9, to identify at least one hospitalization with the diagnosis of endometriosis in the hospital discharge database. In contrast, we used the Systematized Nomenclature of Medicine, SNOMED (codes M-89311, D7-72000, D7-72010, D7-72020, D7-72100, D7-72106, D7-72110, D7-72114, D7-72120, D7-72122, D7-72124, D7-72126, D7-72130, D7-72132, D7-72134, D7-72140, D7-72142, D7-72144, D7-72146, D7-72160, D7-72170, D7-72180, D7-72190), to identify endometriosis in the anatomic pathology database.

We selected women 15–50 years of age residing in the FVG region.

We considered as case a woman with:At least one hospitalization with a diagnosis of endometriosis associated with the identification of endometriosis from the anatomic pathology database. The association was via a unique anonymous identifier of women and temporal proximity.At least one hospitalization with a diagnosis of endometriosis confirmed by laparoscopy or a surgical procedure allowing direct visualization.Identification of endometriosis from anatomic pathology database, without hospitalization with a diagnosis of endometriosis.

Patients with diagnoses supported by imaging procedures alone (i.e., ultrasound, magnetic resonance, computerized axial tomography) were excluded.

To identify incident cases, we included only patients without a diagnosis of endometriosis, based on the criteria described above, in the previous 10 years.

We calculated for each calendar year incidence rates with the number of endometriosis cases in the age range 15 to 50 as the numerator and the number of women aged 15 to 50 residing in FVG, derived from the Italian National Institute of Statistics, [28] as the population-time denominator. We then stratified the incidence rates by age classes, classifying cases based on their age at diagnosis in 15–20, 21–25, 26–30, 31–35, 36–40, 41–45, 46–50.

The analyses were performed using SAS software, Version 9.4 (SAS Institute Inc., Cary, NC, USA).

### 2.2. Standardized Incidence Ratios

Following internal indirect standardization and classification of the population in 7 age classes (15–20, 21–25, 26–30, 31–35, 36–40, 41–45, 46–50), a set of reference rates (Friuli Venezia Giulia region, 2004–2017) were used to compute the expected number of cases for each municipality.

For each i-th municipality (i = 1, …, 216), we calculated the standardized incidence ratio (SIR = observed / expected number of cases) as an estimate of the relative risk (RR), i.e., the disease risk in each area compared to the adopted standard. In this way, we implicitly specified the same null hypothesis H_0_: RR_i_ = 1, testing the procedure for each area through indirect standardization [29].

## 3. Methods

### 3.1. Models for Disease Mapping

We used the two most common Bayesian hierarchical models for Disease Mapping: the Poisson-Gamma model [30] and the Besag-York-Mollié (hereafter known as BYM) model [31]. The objective of these models is to account for overdispersion and stabilize relative risk estimates.

In detail, let’s assume that the observed number of incident cases of endometriosis in the i-th municipality Y_i_ (i = 1, …, 216) follows a Poisson distribution with mean E_i_ x θ_i_, where E_i_ is the expected number of cases under indirect standardization and θ_i_, is the relative risk. The maximum likelihood estimator of θ_i_ is called the standardized incidence ratio (SIR: ${\hat{\vartheta }}_{i}=Y_{i}/E_{i}$).

Bayesian inference requires the specification of appropriate prior distributions on model parameters.

Clayton and Kaldor [30] assumed a Gamma (k,ν) prior distribution for θ_i_. The hyperparameters k and ν are assumed to be exponentially distributed. Poisson random variability is filtered out in this model, and relative risk estimates are shrunken toward the general mean.

Besag et al. [31] specified a random effect log-linear model for the relative risk log(θ_i_) = u_i_ + v_i_. The heterogeneity random term u_i_ represents an unstructured spatial variability component assumed a priori distributed as Normal (0, λ_u_).

The clustering random term v_i_ represents the structured spatial variability component assumed to follow a priori an intrinsic conditional autoregressive (ICAR) model. In other words, denoting S_i_ as the set of the areas adjacent to the i-th one, v_i_|v_j__∈Si_ is assumed distributed as Normal(${\bar{v}}_{i}$, λ_v_ n_i_) where ${\bar{v}}_{i}$ is the mean of the terms of areas adjacent to the i-th one [32] and λ_v_ n_i_ is the precision which is dependent on n_i_, the number of areas in S_i_. The hyperprior distributions of the precision parameters *λ_v_*, *λ_u_* are assumed to be Gamma (*0.5,0.0005*) [33].

The BYM model shrinks the relative risk estimates both toward the local and the general mean through these two random terms.

### 3.2. Profiling Municipalities at High Risk

Identifying the municipalities at high risk leads us in a multiple comparison framework since we have m test of hypothesis (m = 216, number of municipalities in the region). We control for uncertainty due to multiplicity.

#### 3.2.1. Simple Approaches Based on *p*-values

The commonly controlled quantity to account for multiple testing is the Family Wise Error Rate (FWER), and the most common method is the Bonferroni approach. Benjamini e Hochberg [34] proposed a procedure to control the proportion of false rejections among the total number of rejections. They called this quantity the False Discovery Rate (FDR), which is more appropriate in our context since we are performing m identical tests with m different implications if the null hypothesis would be rejected [22]. Storey [20] proposed an exploratory use of the positive FDR (pFDR), i.e., the FDR conditional to having at least one rejection. The pFDR can be interpreted as a posterior Bayesian probability and can be used to define the q-value, Prob (H_0_ | T ≥ T_obs_), T being a test statistic, and T_obs_ the observed value, for a generic i-th test. The q-value is the minimum pFDR in which we can incur in the rejection of the null hypothesis based on the observed or more extreme values of the test statistics. The q-value is a measure that takes multiple testing into account. The “frequentist” calculation of q-values is based on ordered *p*-values and is reported in Reference [20]. In [20], an empirical Bayesian procedure is proposed. When profiling is the study’s goal, as in our case, it can be useful to screen each area using the q-values instead of simply classifying them as “significant/non-significant” according to the assumed level of FDR that is being controlled [35].

For a single hypothesis test, Goodman [36] proposed using the minimum Bayes factor to evaluate how far the observed data moves us from an initial null state. Briefly, if we consider two hypotheses H_0_ and H_1,_ and the data Y, the Bayes theorem implies
$$\frac{P(H0|Y)}{P(H1|Y)}=\frac{P(Y|H0)}{P(Y|H1)}\frac{P\left(H0\right)}{P\left(H1\right)}$$
where $\frac{P\left(H0\right)}{P\left(H1\right)}$ is the prior Odds, $\frac{P(Y|H0)}{P(Y|H1)}$ is the Bayes Factor (BF) and $\frac{P(H0|Y)}{P(H1|Y)}$ is the posterior Odds. The BF quantifies the evidence of data Y for H_0_ vs. H_1_.

The minimum Bayes factor is the smallest possible Bayes factor for the point null hypothesis against the alternative within the specified class of alternatives.

Edwards et al. [37] show that if H_0_: µ = µ_0_, Y~Normal (µ,σ^2^) then BF =$\;\frac{P\left(Y|H0\right)}{P(Y|H1)}\geq \exp\left(-0.5t^{2}\right)$ where t = (y − µ_0_)/ σ is the number of standard errors from the null value. Sellke et al. [38] proposed an approach that works directly with *p* values.

For our analysis, we used two previously described approaches based on the transformation of *p*-values into Bayesian posterior probabilities of the null. The q-values, when considering the whole set of observations accounting for multiple comparisons, and the posterior probabilities obtained from the minimum Bayes factor under three null prior probabilities (optimistic with odds 1:3, neutral 1:1, pessimistic 3:1).

The analyses were performed using STATA software version 14 (StataCorp, College Station, TX, USA).

#### 3.2.2. Hierarchical Bayesian Mixture Model

When selective inference, aiming to identify divergent areas, is the goal, the Bayesian models used for Disease mapping (Section 3.1) are further complicated by introducing a third level into the hierarchy, assuming a mixture model for the unknown relative risks θ_i_ [27].

The likelihood for Y_i_ is still Poisson (E_i_θ_i_), where E_i_ is the expected number of cases and θ_i_ the relative risk in the i-th municipality. We then assume that log(θ_i_) = r_i_ μ_0i_ + (1-r_i_) μ_1i_ where, i.e., the logarithm of the relative risk θ_i_ is modeled as the mixture of two components: μ_0i_, the value of the log relative risk under the null hypothesis H_0_, and μ_1i_ the corresponding value under the alternative H_1_. The r_i_ indicator denotes the group membership.

Under the null H_0_ we assume that all the probability mass is concentrated at one point, i.e., μ_0i_ = 0, leaving only a Poisson random variability. Under the alternative H_1_ extra Poisson variability, reflecting the heterogeneity of relative risk among areas is modeled according to the Poisson Gamma or the BYM models.

The prior distribution for the indicator of the unknown true status, r_i_, is assumed to be Bernoulli distributed with parameter π_i_, which, in turn, is modeled as Beta (α_1_, α_2_) distribution.

The quantity of interest for each i-th area is some appropriate summary measure over the posterior distribution of π_i_–i.e., the posterior classification probability to belong to the null hypothesis set. The term “classification probabilities” underlines the connection to classification theory and clearly distinguishes it from Prob (θ_i_ >1|Y), denoted as posterior probabilities [39,40].

The a priori distribution for π_i_ is assumed to be an exchangeable informative Beta(α_1_, α_2_). Changing the value of the Beta parameters α_1_, α_2_, we a priori introduced in the model our prior belief on the percentage of divergent areas, consistent with the non-Bayesian simple approaches presented in Section 3.2.1.

All the Bayesian analyses were performed using the WinBugs1.4 software [41].

We ran two independent chains for each model, and the convergence of the algorithm was performed following Reference [42]. We discarded the first 100,000 iterations (burn-in) and stored for estimation 50,000 iterations.

### 3.3. Sensitivity Analysis

To compare areas at higher and lower risk within the region for explanatory variables, we planned to use the Chi-square test for categorical variables and the non-parametric Wilcoxon–Mann–Whitney test for continuous variables (data shown in Appendix A).

We used capture-recapture methods to evaluate the coverage of the different registries (hospital discharges and anatomic pathology) and the presence of possible ascertainment bias [43].

## 4. Results

For the period 2004–2017, 4125 new cases of endometriosis were identified in the age range 15 to 50 (Table 1). The crude incidence rate (×100 000) of endometriosis in women aged 15–50 years for the period 2004–2017 was 111, if we consider all diagnoses, with and without histological confirmation (Table 1). The age-specific incidence rate of endometriosis was highest in the age group 31–35 (160 × 100 000).

Of the 4125 cases, 1471 (35.7%) were hospitalizations with a diagnosis of endometriosis associated with the identification of endometriosis from the anatomic pathology database. Of this, 1846 (44.8%) were hospitalizations with a diagnosis of endometriosis confirmed by laparoscopy or other surgical procedure allowing direct visualization, and 808 (19.6%) were identified only from the anatomic pathology database.

### 4.1. Disease Mapping

The SIRs in the period 2004–2017 range between 0 and 3.5 (Figure 1A). The geographical distribution is heterogeneous (Figure 1B) with high/low risk areas. The spatial pattern was more evident if we considered the smoothed map of SIR under the two Bayesian models (Poisson-Gamma and BYM) (Figure 2A,B). The shrinking effect of the Bayesian estimators was evident when comparing it to the SIR map (Figure 1B). Relative risks (RR) of areas with few expected counts and extreme SIR were regressed to the mean. Since we used indirect standardization with FVG reference rates, the observed mean was close to the null RR of one. In the BYM model, the relative importance of the clustering component compared to the heterogeneity component, i.e., the ratio of variances of the two random terms is 4:1. Overall, the geographical pattern of the incidence of endometriosis showed a very strong spatial structure, with southern areas at higher risk.

### 4.2. Profiling Municipalities at High Risk

In Figure 3A, we report the funnel plot [44], a graph of the effect measure (SIR in our case) against its precision (the precision of the SIR is proportional to E_i_ since the asymptotic variance (SIR) = 1/E_i_). This plot shows that the precision in estimating the risk will increase as the population dimension of the area increases. SIR from small areas will spread widely in the diagram, while the variability will be narrower for more populated areas. If the null hypothesis is true for all areas, the plot would be a symmetric funnel centered on the reference line at an ordinate null value of one.

Figure 3A plotted the municipalities with one-sided *p*-values < 0.05, identified with label 1. Figure 3B shows the empirical distribution of the one-sided *p*-values. If no areas diverged, the *p*-value distribution would be uniform. There is evidence of departure from the null because the relative frequency of small *p*-values is greater than expected under the uniform distribution.

Figure 3C shows the quantile-quantile plot of the complementary log transformation of empirical *p*-values against the null theoretical exponential (1) distribution. The plot was useful for identifying the outlying observations responsible for the departure from the uniform distribution shown in panel 3B.

To screen these divergent areas according to a multiple testing correction, we report in Table 2 the municipalities with one-sided *p*-values < 0.05 and the corresponding q-values.

Of the 216 areas examined, 23 have *p*-values < 0.05. Applying Bonferroni’s correction (with a probability of type I error set at 5%), no areas would be divergent from the null. Using Storey’s q-value, 6, 4, and 2 areas were selected when the threshold was set at 20%, 10%, and 5%, respectively.

The last three columns of Table 2 report the calibrated Goodman *p*-values under three different prior odds $\frac{P\left(H0\right)}{P\left(H1\right)}$ 3:1; 1:1; 1:3. This analysis is appropriate if we assume the point of view of each area separately, considering any multiple comparisons adjustment not pertinent. Even with a high probability of the null compared to the alternative, the posterior probability of the null was less than 5% for five municipalities.

In Figure 4, we report the posterior probabilities Prob (θ_i_ > 1|Y) obtained from the Poisson Gamma (A) and the BYM (B) models. The maps are coherent with the distribution of the RR, with areas at high posterior probabilities located in the southwest part of the region. However, considering that such Posterior probabilities are not adjusted for multiplicity, they are useful to describe the overall spatial risk pattern.

The complex Bayesian tri-level modeling embeds the previous points of view in a unified perspective. In Figure 5, we map posterior inclusion probabilities (i.e., the complement to the posterior classification probabilities Prob (r_i_ = 0 | Y > Y_i_; **Y**) = 1 – Prob (r_i_ = 1 | Y = Y_i_; **Y**) [42] under the mixture Bayesian Poisson-Gamma (A, C) and BYM models (B, D). In the first row, inclusion probabilities were obtained specifying a Beta (2.5,7.5) as a priori for π_i_, that is a Beta with an expected value of 25%, that are sensible for a prior belief of a percentage of divergent areas around 75%. In the second row, we specified a Beta (7.5,2.5) for π_i_, which is a Beta with an expected value of 75%, meaning a prior belief of a percentage of divergent areas around 25%. The results were sensitive to prior choices for π_i_ with the Poisson Gamma model. Under the BYM specification, the results were consistent with those obtained by the simpler approaches, and five areas were consistently identified as divergent.

Appendix A presents, as an appendix to Table 2, the distribution of endometriosis cases and population by age classes.

### 4.3. Sensitivity Analysis

Appendix A shows a comparison between the case of endometriosis identified in the five high-risk municipalities and municipalities identified in the other areas of the region, based on the type of identification source and the limited number of demographic variables available from the data sources used. Overall, women in the five municipalities had a higher frequency of histological diagnosis of endometriosis (73.6% vs. 53.8%, *p* < 0.0001) and higher age at diagnosis. No difference was seen concerning the place of birth.

Comparisons with the regional average showed a negative association between anatomic pathology diagnosis and hospital discharge in the five high-risk municipalities. This suggests an ascertainment bias. Either the reporting was more careful than in the rest of the region, or many more pathology reports with a diagnosis of endometriosis were issued without accompanying hospital discharge records. Compared to the regional average, the relative risk observed in the five high-risk areas was 1.50. Fixing a coverage probability equal to that of the five high-risk areas for the rest of the region, we had to increment the number of cases for the rest of the region, and the relative risk of the five high-risk areas would change from 1.50 to 1.31 (95% CI 1.13; 1.52), (See Appendix A for details in methods and results).

## 5. Discussion

Our study shows an estimated incidence of endometriosis consistent with those reported in similar registry-based studies, even if comparisons between studies are subject to limitations due to differences in settings and methodologies applied for case identification and definition. Concerning sampling surveys on endometriosis, in our study and in other similar registry-based ones, the operational definition of the incident case could lead to underestimating the number of disease cases [4,11,12,14,45,46].

The present study’s strength relies on having performed a record-linkage among in/outpatients clinical and pathological databases and having conducted a follow-back to retrieve the date of incidence.

Geographical analysis showed a very strong spatial pattern with high-risk areas in the region’s southeast part. When mapping disease risk, the aim is to estimate the risk distribution on a fine geographical resolution. The problem in meeting this goal is small areas are extremely heterogeneous in population denominators, resulting in difficulties in properly controlling for random variability. In spatial statistical literature, the term Disease Mapping refers to a collection of methods proposed to overcome such difficulties and “stabilize” or “smooth” the risk map. All these approaches rely on shrinkage estimators, and, among these, Bayesian estimators appear to be the most interesting [47].

Bayesian approaches to smoothing relative risk estimates may be misinterpreted as a solution to the problem of selective inference and multiple comparisons in Disease Mapping. However, estimation is a different task from testing. Multiple testing corrections based on *p*-values can be used to select high-risk areas. In a full Bayesian perspective classification, probabilities obtained from full Bayesian mixture models are adjusted for multiple comparisons and have the advantage of an easy way to perform sensitivity analysis on model assumptions. Both “frequentist” and full Bayesian analysis confirm a cluster of 5 adjacent municipalities in the FVG region, for a total of 296 cases of endometriosis (SIR 1.5; 90% CI = 1.36; 1.65) and 98 attributable cases (90% CI = 71; 128).

An ascertainment bias cannot be excluded and could be geographically structured, contributing to explain the observed cluster of endometriosis incident cases. Indeed, a higher frequency of histological diagnosis was recorded in the five high-risk municipalities. This finding lends itself to two opposite interpretations. On the one hand, a more serious disease requiring increased diagnostic and surgical invasiveness. On the other hand, heightened attention to the ascertainment of endometriosis by gynecologists or pathologists working in the five areas is required. We used capture-recapture methods to evaluate for the presence of potential ascertainment bias. While our analyses suggest that there is some evidence of ascertainment bias, this is not enough to explain the observed cluster of cases in the five high-risk areas. We can thus conclude that the five areas are indeed at higher risk. Unfortunately, the very limited information available from the data sources used to identify women with endometriosis does not allow for a more detailed description of patients’ exposure and characteristics. The literature identifies several risk factors for endometriosis, including individual factors (i.e., shorter menstrual cycle length, low body mass index, lower parity, skin sensitivity and other somatic characteristics, Asian origin, autoimmune diseases, familiarity, genetics), behavioral factors (i.e., diet, physical activity, alcohol use, caffeine intake), and environmental pollution (i.e., dioxin, PCB, metals, phthalates) [48].

The most important Italian shipyard in the identified high-risk area is part of the largest shipbuilder in Europe. The company builds both commercial and military vessels. The area also comprises an oil and coal-powered energy plant ranking in the highest quartile of Italian energy plants and currently undergoing a heavily contested authorization renewal procedure (See Appendix A, for further information on high-risk area).

In order to adequately address the role of the above-mentioned risk factors, an ad hoc study should be conducted that includes biomonitoring, evaluation of the individual and behavioral characteristics of the population, and life-course analysis of the exposures. Future studies will address these issues to explain the excess cases in these particularly environmentally stressed municipalities of FVG.

## 6. Conclusions

Our study is based on aggregated data at the municipality level. The main limitation of the study is that we cannot exclude a potential ecological bias. The statistical analysis by Bayesian spatially structured random effects should partially control for spatial confounding. The second limitation is that we cannot exclude a residual differential ascertainment bias among geographical units. We conducted a sensitivity analysis depicting a pessimistic scenario in which more attention is given to detecting endometriosis in the high-risk areas identified. Despite adopting these mitigation measures, we should be cautious in interpreting results from any geographical analysis, particularly when we lack information on spatially structured causal factors.

Generally, geographical variability in the occurrence of endometriosis is considered difficult to interpret because of lack of comparability in diagnosis, selection bias, and heterogeneity in study design across studies [49,50,51]. The use of geographic characteristics in the analysis was restricted to factors or characteristics not immediately interpretable as geographic–such as rural/urban areas, exposure to POPs, and so on [52,53]. When performed spatial analysis, results provide limited information due to potential ascertainment bias [13,54]. In our study, geographic analysis was the primary goal. We were especially careful to minimize differential ascertainment bias across geographical units and adopted sophisticated statistical methods to deal with multiple comparisons and selective inference. Spatial analysis and profiling of high-risk areas are useful tools to address environmental hypotheses in registry-based epidemiological studies.

## Figures and Tables

**Figure 1 ijerph-18-07175-f001:**
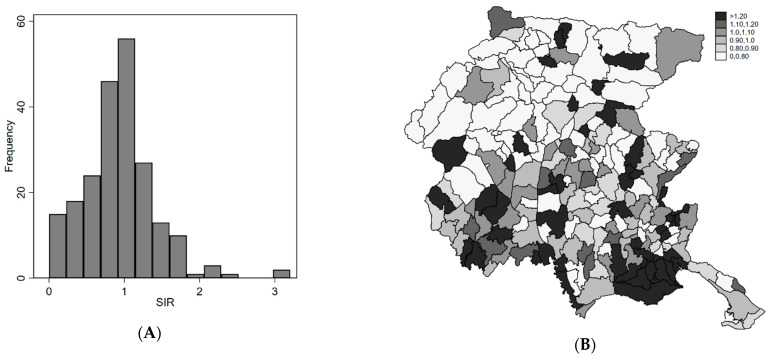
(**A**) Histogram of SIRs (**B**) Spatial distribution of SIRs. FVG, 2004–2017.

**Figure 2 ijerph-18-07175-f002:**
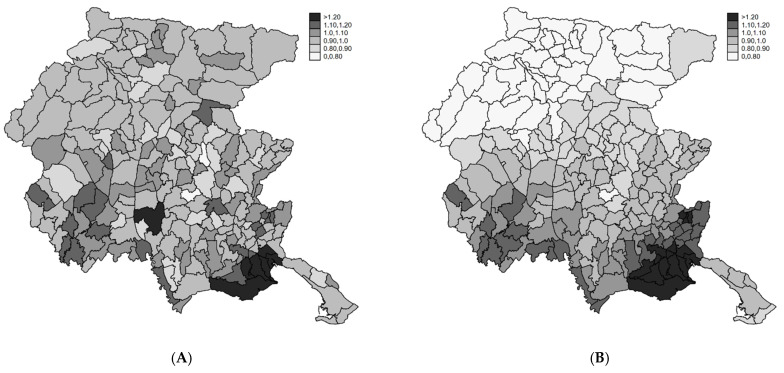
Posterior relative risk estimates from the Poisson-Gamma (**A**) and BYM (**B**) models (see text). FVG, 2004–2017.

**Figure 3 ijerph-18-07175-f003:**
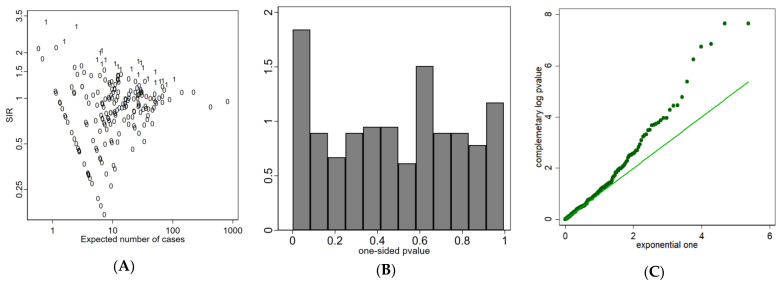
Funnel plot of SIRs: label 1 indicates areas with one-sided *p*-value < 0.05 (**A**); Histogram of empirical one-sided *p*-values * (**B**); Quantile–Quantile plot of complementary log empirical *p*-values versus theoretical exponential (1) (**C**). FVG, 2004–2017. * For each area out of m, the one-sided *p*-value Prob (Y **≥** Yobs|H0) under the null hypothesis H0: θ = 1 against the alternative H1: RR > 1 is obtained from the exact Poisson distribution.

**Figure 4 ijerph-18-07175-f004:**
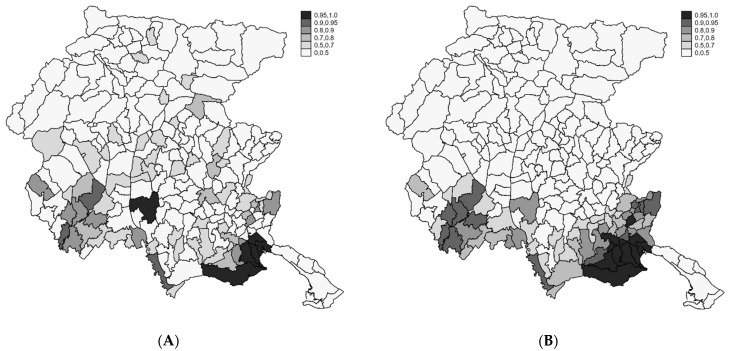
The posterior probability of RR > 1 from the Poisson-Gamma (**A**) and BYM (**B**) models. FVG, 2004–2017.

**Figure 5 ijerph-18-07175-f005:**
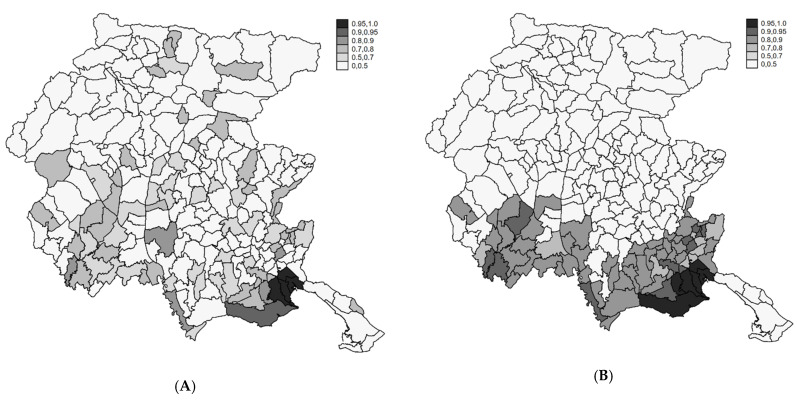
Posterior inclusion probabilities under the tri-level Poisson-Gamma under a priori P(H0): 25% (**A**); Posterior inclusion probabilities under the tri-level BYM models under a priori P(H0): 25% (**B**); Posterior inclusion probabilities under the tri-level Poisson-Gamma under a priori P(H0): 75% (**C**); Posterior inclusion probabilities under the tri-level BYM models under a priori P(H0): 75% (**D**). Endometriosis, FVG, 2004–2017.

**Table 1 ijerph-18-07175-t001:** Age-specific incidence of endometriosis in women residing in FVG in the years 2004–2017.

Age	Women Residing in the Region *	Endometriosis n (Rates × 10^5^)
15–20	402,526	48 (12)
21–25	365,815	228 (62)
26–30	442,259	631 (143)
31–35	541,962	869 (160)
36–40	633,878	846 (133)
41–45	681,123	833 (122)
46–50	654,892	670 (102)
total 15–50	3,722,455	4125 (111)

* Numbers represent the sum of women residing in the region in the fourteen years considered.

**Table 2 ijerph-18-07175-t002:** Municipalities with one-sided *p*-value < 0.05, number of cases, SIR, *p*-value, FDR (q-value), calibrated Goodman *p*-values under 3 alternative prior odds between the null and alternative hypothesis (3:1; 1:1; 1:3). FVG 2004–2017.

					Decrease in Probability of the Null Hypothesis		
Municipality	Number of Cases	SIR ^a^	*p*-Value	q-Value	From 75% to No Less than	From 50% to No Less than	From 25% to No Less than
San Canzian d’Isonzo	38	1.759	0.0005	0.0522	0.0127	0.0043	0.0014
Staranzano	41	1.720	0.0005	0.0522	0.0128	0.0043	0.0014
Ronchi dei Legionari	62	1.498	0.0011	0.0640	0.0260	0.0088	0.0030
Monfalcone	114	1.339	0.0012	0.0640	0.0287	0.0098	0.0033
Grado	41	1.595	0.0019	0.0836	0.0442	0.0152	0.0051
Lusevera	6	2.986	0.0046	0.1672	0.0924	0.0328	0.0112
San Lorenzo Isontino	11	2.073	0.0085	0.2619	0.1480	0.0547	0.0189
Fiumicello	26	1.565	0.0117	0.2858	0.1864	0.0709	0.0248
Codroipo	72	1.307	0.0119	0.2858	0.1892	0.0722	0.0253
Mariano del Friuli	10	1.998	0.0138	0.2982	0.2096	0.0812	0.0286
Morsano al Tagliamento	15	1.697	0.0190	0.3253	0.2587	0.1042	0.0373
Latisana	64	1.295	0.0191	0.3253	0.2593	0.1045	0.0374
Gradisca d’Isonzo	31	1.438	0.0209	0.3253	0.2743	0.1119	0.0403
Capriva del Friuli	11	1.798	0.0229	0.3253	0.2900	0.1198	0.0434
Turriaco	16	1.624	0.0239	0.3253	0.2977	0.1238	0.0450
Prata di Pordenone	42	1.350	0.0249	0.3253	0.3047	0.1275	0.0464
Barcis	2	3.203	0.0256	0.3253	0.3096	0.1300	0.0475
Polcenigo	17	1.557	0.0303	0.3528	0.3401	0.1466	0.0542
Cordenons	78	1.233	0.0310	0.3528	0.3448	0.1492	0.0552
Fiume Veneto	53	1.276	0.0359	0.3865	0.3726	0.1652	0.0619
Arba	8	1.801	0.0376	0.3865	0.3811	0.1703	0.0640
Dolegna del Collio	3	2.378	0.0394	0.3865	0.3901	0.1758	0.0664
Mossa	9	1.689	0.0454	0.4263	0.4179	0.1931	0.0739

^a^ SIR = Standardized Incidence Ratio.

## Data Availability

The data underlying this article were provided by the FVG Regional Health Authority. Data will be shared on request to the corresponding author with the permission of the FVG Regional Health Authority.

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
