# Peer review of "Spatial Patterns of Endometriosis Incidence. A Study in Friuli Venezia Giulia (Italy) in the Period 2004–2017"

_ijerph, 2021, doi:10.3390/ijerph18137175_

Round 1

Reviewer 1 Report

This article calculated the geographical distribution of incidence of endometriosis by sophisticated statistical methods. They pointed out the high risk area in Friuli Venezia Giulia of Italy. This article contains very important issues. There is still a question to which authors should answer.

  1. Authors found that southern area of FVG is high risk for endometriosis. Authors are advised to describe the characteristics of the southern area of FVG, i.e. rural area, industrial area, etc…, because most readers are unfamiliar with Italy.
  2. Authors should describe the geographical characteristics of the areas they studied. How were economics, climate, industry, and so on, in the high risk area of endometriosis?
  3. If the data of age distribution in each area are presented, it would add more valuable information. In table 2, the data of age distribution should be supplemented.

Reviewer 2 Report

Review ijerph-1236946 - Geographical distribution of incidence of endometriosis in Friuli Venezia Giulia (Italy) in the period 2004-2017

The article deals with a very important and valuable topic concerning the epidemiology of endometriosis cases. Currently, there are many uncertainties on this issue.

I would like to commend the authors for a very well-prepared work that brings interesting insights into the epidemiology of endometriosis. Although the geographic region is limited, the authors' conclusions may apply to other territories.

Manuscript is well written, the discussion is interesting, and the selection of references is up-to-date and well-chosen.

I would like to commend the authors for a  well-prepared work that brings interesting insights into topic but manuscript needs some changes to solve following remarks:  

  • Title  - should be changed for more interesting for a wider readership  
  • - The discussion  - needs rebuilding:  -  should be more developed, especially in terms of comparison to other similar studies - there are only 2-3 vague sentences  - This will   highlight the value of the current research   - This section of the manuscript lacks considerations that are of interest to non-narrow geographic readers and conclusions that apply to other      areas.  
  • - The paragraph on the limitations of the current study should also be clearly emphasized.

Reviewer 3 Report

Diagnosing endometriosis depends on the assessment methods and the gynecologist, as well. It should be mentioned in introduction section.

Round 2

Reviewer 1 Report

This article has been revised well with the appropriate response to the reviewer’s comments. Now it would be suitable for publication in the journal.

Author Response

Thank you for your kind comment and for the opportunity to improve our manuscript.

Reviewer 2 Report

Review of revised version of manuscript: ijerph-1236946

The changes introduced by the authors significantly improve the quality of the manuscript.
I have 2 more comments and suggestions for changes:
  - I do not understand the part of the sentence in the Introduction: As observed by a reviewer, diagnosing endometriosis ..... - Is it based on the observations of the reviewers (including my observation?) during the preparation of the manuscript? - If so then it is wrong - the review process is not part of the manuscript content - it needs to be corrected
  - the newly added part about work restrictions is well described but should be included in the discussion at the end before the final conclusions
